

# Probabilistic short-range forecasts of high precipitation events : optimal decision thresholds and predictability limits

François Bouttier[1] and Hugo Marchal[1]

[1]CNRM, Université de Toulouse, Météo-France, CNRS, Toulouse, France

**Correspondence:** François Bouttier (francois.bouttier@meteo.fr)

**Abstract.**

Translation of ensemble predictions into high precipitation warnings is assessed using user oriented metrics. Short range probabilistic forecasts are derived from an operational ensemble prediction system using neighbourhood post-processing and conversion into categorical predictions by decision threshold optimization. Forecast skill is modelled for two different types of

users. We investigate the balance between false alarms and missed events and the implications of the scales at which forecast information is communicated.

Results show that ensemble predictions objectively outperform the corresponding deterministic control forecasts at low precipitation intensities when an optimal probability threshold is used. Thresholds estimated from a short forecast archive are robust with respect to forecast range and season and can be extrapolated towards extreme values to estimate severe weather

guidance.

Numerical weather forecast value is found to be limited: the highest usable precipitation intensities have return periods of a few years only, with resolution limited to several tens of kilometres. Implied precipitation warnings fall short of common skill requirements for high impact weather, confirming the importance of human expertise, nowcasting information and the potential of machine learning approaches.

The verification methodology presented here provides a benchmark for high precipitation forecasts, based on metrics that are relatively easy to compute and explain to non-experts.

## 1 Introduction

Numerical weather prediction increasingly relies on ensemble prediction systems that estimate probabilities of atmospheric events. Ensemble members are computed by numerical integrations of numerical models in order to inform the interpretation

of model output in the presence of predictive uncertainties. An important application is the issuance of high precipitation warnings which can be useful for protection against flooding and inundation events causing losses of human life and economic and environmental damage (WMO, 2021).

This work presents a general method for computing high precipitation forecasts and for measuring their skill. It is tested with a state-of-the-art ensemble numerical prediction system. The dataset used here is relevant for short-range forecasts (for the next

few hours) over western Europe. We aim to clarify the conditions under which threshold exceedance for accumulated precipi-



tation can be skilfully predicted as an input to severe weather warnings. Previous studies have demonstrated that probabilistic forecasts derived from ensembles contain potentially valuable predictive information, but using this information in practical applications is often hampered by challenges in communicating and interpreting probabilities (Joslyn and Savelli, 2010; Fundel et al, 2019; Demuth et al, 2020) This problem is acute for predicting flash floods and inundations, because precipitation

uncertainties can impact hydrological forecasts, either as drivers of rainfall–runoff models, or when they are directly used to estimate flood risks (Cloke and Pappenberger, 2009; Ramos et al, 2010; Hapuarachchi et al, 2011; Zanchetta and Coulibaly, 2020).

Limits to the usefulness of ensemble prediction were explored by Buizza and Leutbecher (2015) who defined the forecast horizon as the range at which the CRPS (Continuous Ranked Probabilistic Score) of the ECMWF ensemble system ceases to be

statistically better than climatology. In terms of upper level atmospheric variables, they found that the horizon is several weeks long and depends on the considered horizontal scales, which were of the order of 100–1000 km in their work. Their approach seems difficult to apply to short range predictions of severe, high-impact weather events that tend to have much smaller scale. Another issue is that the practical significance ot skill measures such as the CRPS can be hard to understand for non-expert users.

Here, we focus on high precipitation events that are conducive to flash floods, and on the ability of numerical weather prediction to forecast them with enough anticipation and precision to be useful for issuing severe weather warnings. Many case studies have demonstrated that current numerical prediction tools have limited skill in predicting high precipitation and flash floods: a common problem is a lack of precision of numerical forecasts of precipitation, as illustrated in articles by Golding et al (2005); Vié et al (2012); Davolio et al (2013); Zhang (2018); Martinaitis et al (2020); Sayama et al (2020); Caumont et

al (2021); Furnari et al (2020); Amengual et al (2021); Godet et al (2023), among others. Precipitation forecast uncertainties are not the only source of errors in flash flood prediction, but they are widely regarded as a major issue that has motivated the development of hydrometeorological ensemble prediction systems. The hope that a detailed representation of precipitation forecast uncertainties will improve estimates of flash flood and inundation risks, as discussed by Collier (2007); Cloke and Pappenberger (2009); Dietrich et al (2009); Demeritt et al (2010); Hapuarachchi et al (2011); Zappa et al (2011); Addor et

al (2011); Alfieri et al (2012); Demargne et al (2014). Now that high resolution atmospheric and hydrological ensemble prediction systems are running operationally in several centres, it is interesting to check the relevance of precipitation forecasts in terms of specific user needs.

Modern regional atmospheric ensemble prediction systems have the appropriate grid resolution and timeliness to simulate high precipitation events. They use convection permitting numerical atmospheric models at kilometric resolutions. Several

studies have shown that they have skill in predicting localized, intense phenomena such as thunderstorms, tornadoes and high precipitation, when compared to lower resolution model systems (e.g., Stensrud et al, 2013; Clark et al, 2016; Zhang, 2018). Published verification studies of convection permitting ensembles, however, tend to rely on few case studies or precipitation scores at limited intensities.

Here, we aim to quantify the objective value of ensemble-based forecasts of intense precipitation, for users interested in

categorical "yes/no" forecasts that future precipitation will exceed predefined intensity thresholds. The originality of this study





is that (a) we score short range precipitation forecasts with the highest possible intensity given a training dataset of one year of ensemble predictions, (b) we focus on performance measures that are most likely to be relevant to forecast end users, (c) we present an approach that can be applied to generate useful end user products from ensemble predictions.

The existing literature suggests that these objectives require dealing with three problems : defining scores that are both meaningful to non-experts and appropriate for rare events, designing an effective neighbourhood verification framework to deal with the "double penalty" issue as explained below, and achieving a meaningful balance between event detection and false alarm rates in the forecast products. Solutions will be presented and discussed in the following.

A key issue is the inherent rarity of high impact precipitation, which makes it difficult to produce statistical evidence of the value of intense precipitation forecasts. Verification of probabilistic predictions is tricky for rare events because the low climatological frequency (or base rate) of the events implies that several commonly used scores, such as the area under the relative operating characteristic (ROC) curve, or the critical success indicator (CSI), exhibit an excessive sensitivity to hit rates at the expense of false alarms. This problem was recognized as early as Doswell et al (1990). Attempts to solve this issue (Sharpe et al, 2018; Yussouf and Knopfmeier, 2019) relied on the definition of more appropriate scores, such as the weighted CRPS, the fractions skill score (FSS), and the Symmetric Extremal Dependency Index (SEDI, Ferro and Stephenson (2011)). These scores have attractive theoretical properties such as "being proper" which make them useful for developers of operational forecast systems because they allow clean comparisons between different systems. Unfortunately they can be difficult to compute and difficult to understand for non experts, particularly in terms of absolute predictive performance. We argue that, for end users to effectively use ensemble and probabilistic forecasts products, they should have access to clearly understandable performance statistics, as advocated by e.g. Joslyn and Savelli (2010); Ramos et al (2010); Fundel et al (2019); Demuth et al (2020). We will focus on two scores that are easily interpreted using concepts of detection or false alarm rates: the equitable threat score (ETS), which is well known in the weather forecasting community (Jolliffe and Stephenson, 2011), and the F2 score, which is standard in the machine learning community (Wikipedia, 2023) and has desirable properties with respect to rare events, as explained later.

Another issue is the sensitivity of precipitation scores to the space and time scales at which products are used. Ideally, one should focus on scales that are meaningful to the end users. The location and timing errors of extreme precipitation events tend to be similar to, if not larger than the size and duration of the events themselves. Thus, grid scale performance measures of convection permitting models (at ≈1 km scale) will indicate that numerical forecasts have no predictive skill if there is no overlap between forecasts and observed events, a problem called the "double penalty effect". To circumvent it, performance measures can be applied to forecasts that have been upscaled by neighbourhood post-processing in order to assess their skill at coarser resolutions than 1 km. In our study, we will focus on prediction scales of the order of 30 km, which is the resolution at which weather warnings are often issued in European countries. We shall demonstrate that this resolution also makes sense because it is close to the smallest scales at which current numerical precipitation forecasts have significant skill. The design and implications of ensemble neighbourhood postprocessing have been extensively discussed in Ben Bouallègue and Theis (2014) and Schwartz and Sobash (2017). We will use one of their techniques (the neighbourhood maximum ensemble postprocessing,





or NMEP) in this work, because it is relevant to a frequent objective of high precipitation warnings: forecasting whether precipitation is likely to exceed a given threshold, anywhere within a neighbourhood.

A third issue is the balance between forecast misses and false alarms. Most traditional scores for binary events give similar weights to non-detections and false alarms (e.g., the ETS, CSI, HSS, or TSS in Jolliffe and Stephenson (2011)). When predicting extreme events, the consequences of an event miss (i.e., a non-detection) tend to be more serious than a false alarm, because they are high impact events for which the cost of protection can be much lower than the cost of being caught unprepared. Accordingly, warning practice in weather forecasting offices often seeks a ratio of false alarms to non-detections that is much higher than one (it was of the order of 6 in Météo France around 2020; in Hitchens et al (2013), warnings of the USA NOAA/NWS Storm Prediction Center were estimated to have a ratio of 3 to 5). Although typically greater than one, this ratio should not be too high, either, if one wants to avoid eroding the credibility of future warnings by issuing false alarms too often (Roulston and Smith, 2004).

The economic value score (Zhu et al, 2002) accounts for this balance issue in the verification of probabilistic forecasts, using a simple model of the user cost–loss function. This score is not fully satisfactory for interpretation by non-expert users because it requires them to select a cost–loss ratio. Besides, the economic value score indicates a potential value which can only be realized by optimally thresholding forecast probabilities into yes/no forecasts: this operation must be based on statistical analysis of an archive of past forecasts. In our study we will focus on users that seek forecast information directly suitable for decision making. The specific aim is to investigate how probabilities of precipitation threshold exceedance can be converted into categorical exceedance forecasts.

We shall demonstrate that optimal deterministic forecasting rules can be precomputed to account for those three criteria (event rarity, scale selection and balance between detection and false alarms), using a statistical optimization of the ETS or F2 score over past events. The optimization depends on the user preferences regarding the targeted detection to false alarm ratio. These rules can be used to plot easily interpretable ensemble predictions products such as tailored maps of upscaled probabilities or quantiles, with full knowledge of their performance (bias, detection and false alarm rates).

In summary, this paper will present a practical technique for summarizing ensemble predictions of high precipitation in an explainable way, taking into account the preferences of typical users. As a by-product, we will document the predictability limits of the prediction ensemble system in terms of the maximum predictable intensity, and the finest predictable scale.

The following will be structured as follows. In Sect. 2, the dataset (model and observations) is described, and Sect. 3 presents the forecast rule optimization technique. Sections 4, 5 and 6 document the main sensitivities of the optimal rule and forecast performance to the implementation choices. Section 6 presents a few illustrative case studies before the final summary and discussion in Sect. 7.



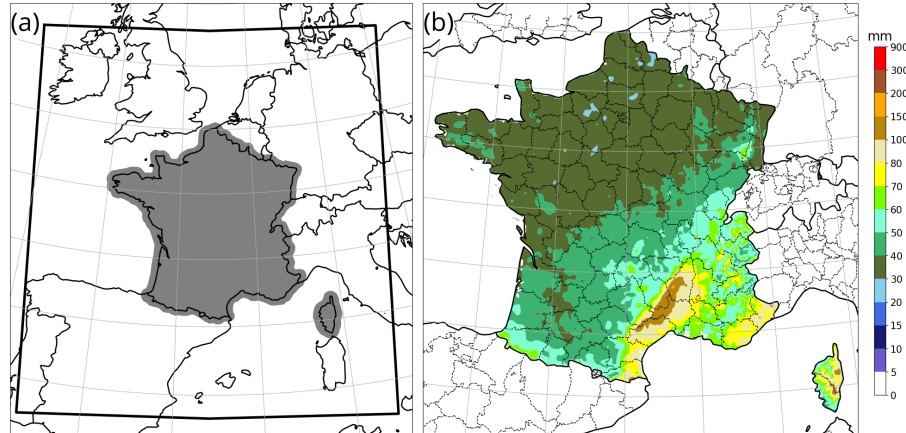

**Figure 1.** (a) AROME computational model domain (black line) and verification area (grey shading), (b) 6-hourly climatological precipitation quantile corresponding to a 5 year return period according to the SHYREG climatology.

## 2 Data and methods

### 2.1 Numerical prediction system

This study is based on an archive of probabilistic forecasts of total precipitation (surface accumulations of rain, snow and graupel) from the operational AROME-EPS ensemble prediction system deployed over Europe by Météo-France. Accumulations are considered over 6 h (unless otherwise mentioned) and indexed by the end time of the accumulation interval. The ensemble contains 17 members : an unperturbed control run called "deterministic AROME" and 16 perturbed members. The numerical prediction model is AROME configured over western Europe (Fig. 1a) with 1.3 km horizontal resolution and 90 vertical discretization levels. AROME is a convection-permitting atmospheric model with non-hydrostatic dynamics and physical parametrizations of cloud and precipitation microphysics, radiation, subgrid turbulence, shallow convection and surface interactions as described in Seity et al (2011) and Termonia et al (2018). The deterministic AROME forecasts start from the analysis of a 3D variational data assimilation system (Brousseau et al, 2011). Its large scale boundary conditions are taken from the global ARPEGE prediction system of Météo France.

Perturbed AROME-EPS members start from the deterministic initial state to which perturbations from an ensemble data assimilation system are added. They also use large-scale and upper boundary conditions from the global PEARP ensemble prediction system of Météo France, perturbed surface conditions as documented in Bouttier et al (2016), and forecast equations modified by a stochastic perturbation (SPPT) scheme (Bouttier et al, 2012).

The dataset used in this paper is a subset of the operational productions. It covers the period from July 2022 to September 2023, with one 17 member ensemble per day issued from the 21UTC analysis and available for use about 2 h after analysis time. Unless otherwise mentioned, the forecast ranges considered are 9 and 21 h after analysis. They are spaced by 12 h in order to minimize effects from the diurnal cycle. Each accumulated precipitation forecast field is interpolated by a nearest



neighbour algorithm to a regular 0.025×0.025 degree latitude–longitude grid. The verifying available observations (described in the next section) are located well inside the model computational domain.

## 2.2   Observations and neighbourhood post-processing

Precipitation forecasts are verified against a truth provided by the 1 km resolution hourly ANTILOPE precipitation analysis (Laurantin, 2013). ANTILOPE merges accumulated precipitation estimated from radar reflectivity observations (Tabary,

2007; Tabary et al, 2013) with raingauge data. Verification statistics are computed in the area depicted in Fig. 1a, where the ANTILOPE analysis quality is deemed similar to raingauge measurements because it benefits from good radar coverage and relatively dense in situ reports. The climatology of high precipitation is not homogeneous over the verification area: as shown by the map in Fig. 1b, the heaviest precipitation events tend to occur in the southeastern Mediterranean region. This map was extracted from the SHYREG climatology (Arnaud et al, 2008), which blends an interpolation of raingauge observations with a

statistical model of extreme values.

The ANTILOPE fields are thinned to 10 km resolution in order to produce pseudo-observations with minimal error correlations between neighbouring reports. It leads to approximately 6800 observation points, i.e., the dataset comprises 60 million model–observation pairs.

Previous studies of precipitation forecast performance have applied bias correction to alleviate systematic errors in model

output, for instance using probability matching (Clark, 2017, e.g.,). Biases in the high precipitation forecasts have been diagnosed in terms of the relationship between the highest observed and forecast quantiles in Fig.2, which shows that relative biases are under 20% up to 35 mm accumulations. Above, the biases are larger, but difficult to remove because they are linked to increasingly rare events. The extrapolation of statistics from moderate to extreme intensities will be further discussed in Sect. 4.

Probabilistic forecasts will be assessed after filtering at several spatial resolutions, which was shown to have a large impact by Ben Bouallègue and Theis (2014) and Schwartz and Sobash (2017). Severe weather warnings are usually issued for areas with substantial spatial extent, as in Legg and Mylne (2004). Two spatial aspects of verification will be taken into account in the following score computations:

   – forecast neighbourhood : ensemble forecasts members are post-processed by a "max-neighbourhood" operator, that is,

the forecast at each verification point is replaced by the field maximum in a disc of radius $R_f$ centred on it. Forecast probabilities computed at each point of this post-processed ensemble were called NMAP ("neighbouring maximum ensemble probabilities") in Schwartz and Sobash (2017). This operation blurs small-scale detail from the model output while preserving local precipitation maxima. A similar technique was used in Bouttier and Marchal (2020) and shown to be effective in extracting skillful information from ensembles of fields with very small scale features.

– verification neighbourhood : the NMEP probabilities at each verification point are compared with ANTILOPE observations to which the max-neighbourhood operator is applied with a radius $R_o$ that may differ from $R_f$. The purpose





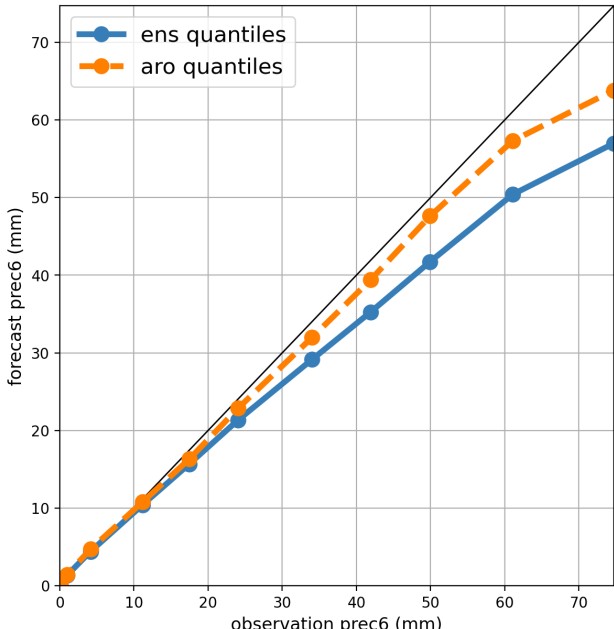

**Figure 2.** Quantile–quantile plot of the AROME ensemble (ens) and AROME control member (aro) precipitation forecast vs ANTILOPE observations, over June 2023.

of observation filtering is to introduce spatial tolerance when comparing forecasts to observations. This procedure was called "upscaled verification" in Ben Bouallègue and Theis (2014).

As recommended by Schwartz and Sobash (2017), we will set $R_f = R_o$ in most of this paper, except when investigating the potential of relaxing this constraint in Sect. 4.

### 2.3  Definition of performance scores

The value of 6-hourly precipitation forecasts will be measured by user-oriented objective scores designed to represent the needs of two hypothetical users :

- User L (for "low") is interested in threshold exceedance of low to moderate precipitation intensities, at the highest possible spatial resolution, and wishes to avoid non-detections as much as false alarms. We will characterize the corresponding forecast performance using the equitable threat score (ETS) with an intensity threshold of 4 mm.

- User H (for "high") is interested in the occurrence of more intense and impactful precipitation, he/she can tolerate more false alarms than non-detections, and aims to issue high precipitation warnings at a resolution of 30 km. User H will consider that a forecast at any given point is satisfactory if the predicted precipitation intensity was observed within a 30 km radius. This objective will be modelled by the F2 score using neighbourhoods $R_f = R_o = 30$ km and an intensity threshold of 30 mm.



The ETS and F2 scores are applied as follows : probabilistic forecasts are generated at each verification point by counting the proportion of members that exceed the chosen intensity threshold. Since the number of ensemble members is small, each discrete set of member values will be converted (or "dressed", Broecker and Smith (2008)) into continuous probability density functions by convolution with a multiplicative triangular kernel of predefined width. When scoring 17 member ensembles, we will set the kernel width so that its standard deviation is 20% of the member value. When scoring deterministic forecasts, we will use a standard deviation of 40%. As reported at the end of Sect. 3, the results presented in this study have little sensitivity to this dressing step, its purpose is to produces smoother score displays than the undressed forecasts.

The aim of this work is to assess the value of deterministic forecast decisions based on ensemble output, a process modelled as a "yes" forecast of high precipitation when the forecast probability of an event of interest exceeds some probability threshold. Note that the probability threshold (also called "operating threshold" in the ROC diagram (Jolliffe and Stephenson, 2011)) should not be confused with the precipitation exceedance threshold (in mm) used to define the forecasted event. As noted by Joslyn et al (2007), Pappenberger et al (2013) and Demuth et al (2020), thresholding forecast probabilities is a common decision making technique when using ensemble forecasts. For the moment, we will treat the probability threshold as a tunable parameter of the forecast procedure. A method to choose its value will be proposed in Sect. 3.

Using the probability threshold, forecast probabilities are converted into deterministic yes/no point forecasts. Their verification against observations leads to the contingency matrix (also called confusion matrix) that counts forecasts successes and failures. For any set of forecast–observation pairs, the counts are computed in terms of the event "precipitation exceeded the intensity threshold" as four categories :

$a$**:** the event was forecast and observed (a hit)

$b$**:** the event was forecast but not observed (a false alarm)

$c$**:** the event was observed but not forecast (a miss)

$d$**:** the event was neither observed nor forecast (a correct negative)

The confusion matrix can be visualized as follows :

|  | event is observed | event is not observed |
| --- | --- | --- |
| event is forecast | a | b |
| event is not forecast | c | d |

Its coefficients are used to define the scores

$$\text{ETS} \quad = \quad (a - a')/(a - a' + b + c) \text{ where } a' = (a+b)(a+c)/(a+b+c+d) \tag{1}$$

$$\text{F2} \quad = \quad a/(a + 1/5b + 4/5c) \tag{2}$$

The ETS gives similar weight to $b$ and $c$, whereas F2 gives 4 times more weight to $b$ than $c$. The F2 score embodies the tolerance to false alarms of User H, with a target ratio of 4 times more false alarms than non-detections. A generalized version of the





F2 score could be used for other target ratios. The confusion matrix can be summarized by various other performance metrics, such as :

**forecast frequency bias** $B = (a + b)/(a + c)$

**hit rate** $HR = a/(a + c)$, also known as "true positive rate" or "recall"

**false discovery rate** $FDR = b/(a + b)$, also known as "probability of false alarm" or "false alarm ratio"

**probability of false detection** $POFD = b/(b + d)$, also known as "fallout" or "false alarm rate"

We shall avoid using the phrases "false alarm rate" and "false alarm ratio" given the confusing surrounding these terms, as explained in Barnes et al (2009).

## 3   Optimization of the decision thresholds

### 3.1   Optimization for User L

Ensemble prediction produces forecast probabilities, which leaves it up to the user to convert them into categorical decisions. Here, we will model a binary decision process by thresholding the probabilities, and we propose to define the threshold $p$, or "decision threshold", as the solution $p_{\text{opt}}$ of an optimization process that maximizes the value of the forecast to the user. To each possible $p$ value, one can associate a set of binary forecasts that leads through comparison with observations to a contingency matrix with coefficients $(a(p), b(p), c(p), d(p))$. The ETS and F2 metrics computed from this matrix can then be regarded as functions of $p$. During rainy events, state of the art numerical weather prediction models normally perform better than the trivial forecasting strategies implied by $p = 0$ (always predict rain) and $p = 1$ (never predict rain). Hence, between 0 and 1 the functions $ETS(p)$ and $F2(p)$ will have a maximum $p_{\text{opt}}$, the optimal decision threshold. The respective values $ETS(p_{\text{opt}})$ and $F2(p_{\text{opt}})$ represent the best value that users can expect to obtain from ensemble forecast output, so they can be regarded as benchmarks of predictive skill. We shall show below that the computation of $p_{\text{opt}}$ is robust enough to be used as a decision tool for various types of users and events, including high precipitation.

The $p_{\text{opt}}$ optimization for a such a forecasting process can be summarized as follows:

1. obtain member fields from the ensemble prediction

2. apply the neighbourhood operator with radius $R_f$ to each field

3. at each point, build the discrete distribution of predicted values

4. apply the dressing operator to each distribution

5. at each verification point, apply the neighbourhood operator with radius $R_o$ to the corresponding observation field



6. for each possible probability threshold $p$ and at each verification point, compute whether the forecast probability of the event exceeds $p$ and increment the confusion matrix coefficients $(a(p), b(p), c(p), d(p))$ according to whether the event
was observed and/or predicted

7. once the matrix has been computed over the whole learning dataset, set $p_{\text{opt}}$ as the $p$ value that maximizes $\text{ETS}(p)$ or $\text{F2}(p)$.

As an example, the computation of $p_{\text{opt}}$ for User L is illustrated in Fig. 3 over June 2023 : the precipitation threshold is set to 4 mm, and the neighbourhood radii $R_f$ and $R_o$ to zero. The computation is performed with two numerical prediction
systems: the 17 member AROME-EPS ensemble, and the deterministic AROME forecast. Both systems are post-processed by the dressing and neighbourhood operator described in Sect. 2. The graphs show various performance diagnostics as a function of the probability threshold $p$, for all $p$ values between zero and one by steps of $0.02$ .

The top panel of Fig.3 shows that, for AROME-EPS, $\text{ETS}(p)$ and $\text{F2}(p)$ are broadly concave curves with different $p_{\text{opt}}$ : $p_{\text{opt}}(\text{F2})$ is lower than $p_{\text{opt}}(\text{ETS})$ which reflects the fact that F2 allows more false alarms than ETS, so that F2-optimal
forecasts of the event will only be issued if the forecast probability exceeds $0.12$ , instead of $0.3$ for ETS-optimal forecasts. The $p_{\text{opt}}$ values are represented in Fig. 3 as vertical lines.

In summary, the best way to warn user L of a risk that precipitation will be above 4 mm is to issue a warning when the forecast probability of this event is above $p_{\text{opt}}(\text{ETS}) = 0.3$ . If, as will be shown later, these optimal decision thresholds are not very sensitive to the choice of precipitation threshold, then the forecast information may be summarized by the 0.7-quantile,
the "optimal quantile", because $0.7 = 1 - 0.3$ .

It is worth noting that the $p_{\text{opt}}$ computation needs not be very precise : given the rounded shape of the $\text{ETS}(p)$ curve, any decision threshold between 0.25 and 0.35 will achieve nearly optimal forecasts. Also, $p_{\text{opt}}$ should not interpreted as a calibrated probability, because the ensemble used is not exactly reliable : $p_{\text{opt}}$ is only a tool to construct an optimal decision rule for the (post-processed but uncalibrated) ensemble output. Some ensemble calibration step could be incorporated into our
precipitation post-processing procedure (e.g., Ben Bouallègue et al (2013), Flowerdew (2014), Scheuerer (2014)). Calibration techniques that simply remap probability values would not improve $\text{ETS}(p_{\text{opt}}$ because it already is optimal in terms of end user scores : our optimization includes an implicit calibration of the ensemble output with respect to the targeted user. More sophisticated calibration techniques could be applied to further increase the value of the ensemble forecast output, in which case our procedure can be applied to the calibrated ensemble instead of the raw members.

The dashed lines in Fig. 1a provide a comparison of the ensemble skill with respect to the deterministic model. The $\text{F2}(p)$ and $\text{ETS}(p)$ curves for the deterministic AROME are nearly flat, except for the trivial forecasting strategies $p = 0$ and $p = 1$. They have a slight slope because of the dressing step. The AROME-EPS scores are higher than the AROME scores, which demonstrates that, even when the end goal is to issue categorical weather predictions, ensemble prediction can provide better forecasts than a deterministic approach if the users know which decision thresholds to use. At non-optimal thresholds, however,
the ensemble scores drop below the deterministic ones. In summary, it is necessary to provide users of ensemble output with supporting information such as $p_{\text{opt}}$ in order to ensure that their forecasts are better than those implied by deterministic models.





Figures 3b and 3c provide additional information about the forecast performance. The HR and FDR curves show that improved detection is obtained at the expense of more frequent false alarms. The ETS optimum achieves nearly equal frequencies of 55% for both kinds of errors, which could be regarded as acceptable for most users of light precipitation forecasts. Lowering the decision threshold would increase detection faster than the false alarms, which is only an improvement for users that favour detection, as modelled by F2. The information conveyed by the HR and FDR curves is similar to a ROC curve (Jolliffe and Stephenson, 2011), with key differences :

- the probability threshold information is hidden in the ROC curve, unless provided by curve labels. Our graphs show its relationship with performance statistics, allowing the user to understand the implications of basing decisions on a particular threshold.

- we illustrate the prevalence of false alarms using the FDR statistic instead of the POFD because the latter collapses to very small values at high precipitation intensities, since the frequency of correct non-event forecasts ($d$ in the confusion matrix) tends to be very large. This occurs because non-occurrence of high precipitation is trivial to predict in a majority of weather situations that are almost certainly dry. Correct non-event forecasts in these cases imply no practical predictive value. In the ROC diagram, large $d$ values lead to a nearly vertical ROC curve that overemphasizes detection skill at the expense of false alarms. This is undesirable, because even when predicting rare events, the frequency of false alarms must be limited to avoid loss of user confidence in the forecast products (Roulston and Smith, 2004). Using FDR instead of POFD avoids this problem.

Figure 3c shows the forecast bias and the ratio of false alarms to non-detections (ratio $b/c$ from the confusion matrix). It indicates that the ETS-optimal decision strategy yields forecasts that overforecast the event by about 20% . It also produces more frequent false alarms that non-detections, which may seem counterintuitive because the ETS score equation puts an equal weight on both kinds of errors. The explanation is that this strategy produces a larger amount of correct forecasts (coefficient $a$) than the threshold that equalizes $b$ and $c$. The graphs show, however, that the strategy that produces bias-free forecasts (with $p = 0.35$) is still rather acceptable because its ETS is only slightly lower than ETS($p_{\text{popt}}$) (by about 5% ).

In summary, we have shown that, according to the June 2023 dataset, the optimal forecast strategy for User L is to set the decision threshold to $p_{\text{opt}}(\text{ETS}) = 0.3$ , which implies overforecasting the event (rr6 > 4 mm) by 20% , with a hit rate and probability of false alarm around 55% .

## 3.2 Optimization for User H

The same methodology is now applied to User H, modelled by a precipitation threshold of 30 mm in 6 h with forecast and verification neighbourhood radii $R_f = R_o = 30$ km. It leads to the plots in Fig. 4. The overall shapes and values of the ETS and F2 score curves are similar to Fig. 3. User H is more tolerant to false alarms, and the optimal decision threshold for AROME-EPS forecasts is $p_{\text{opt}}(\text{F2}) = 0.18$ . Error statistics are worse than for User L : the hit rate drops to 0.56 while the probability of false alarm rises to 0.83 . The forecast bias becomes substantial, since the event is now forecast three times more often than observed. False alarms occur ten times more often than non-detections, which is a consequence of using the F2 score. If one





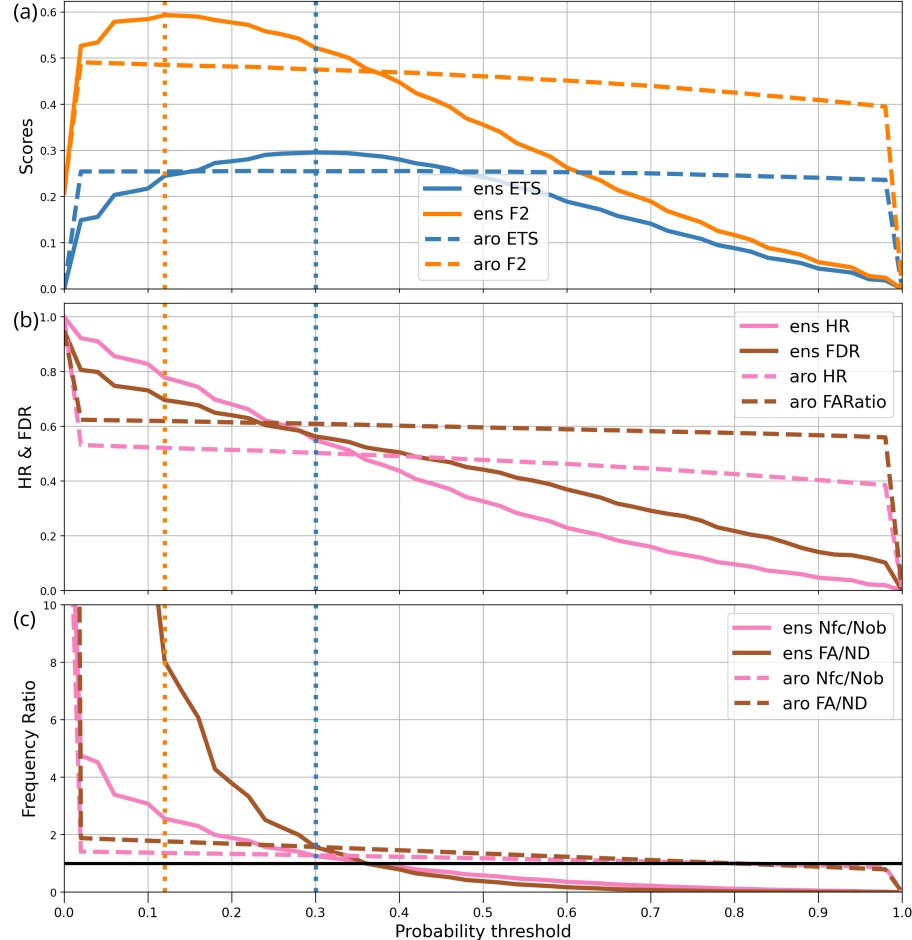

**Figure 3.** Verification statistics of precipitation post-processed over June 2023 for User L, as a function of probability threshold. For the AROME-EPS ensemble (ens) and the AROME (aro) deterministic forecast, the curves show (a) the ETS and F2 scores, (b) the hit rate (HR) and false discovery rate (FDR), (b) the frequency ratios of forecasts to observations (Nfc/Nob) and of false alarms to non-detections (FA/ND). The vertical dotted lines indicate the optimal decision thresholds for the ETS (orange) and F2 (blue) score. The horizontal black line in panel (c) indicates the value 1.

constrains the AROME-EPS-based forecasts to be unbiased, instead, by choosing decision threshold $p = 0.38$, the false alarms are reduced at the price of a large hit rate drop to 0.3. It shows that overforecasting heavy rain events is necessary to achieve F2-optimal forecast performance.

One could argue from Fig. 4 that the deterministic AROME seems better than AROME-EPS, since for $p \simeq 0.35$ its hit rate and F2 score are better, and its bias is lower. Unfortunately, its probabilities of false alarm are also much higher, so the answer depends on the relative importance given to each metric. In terms of the F2 score, AROME-EPS is a better forecasting system.





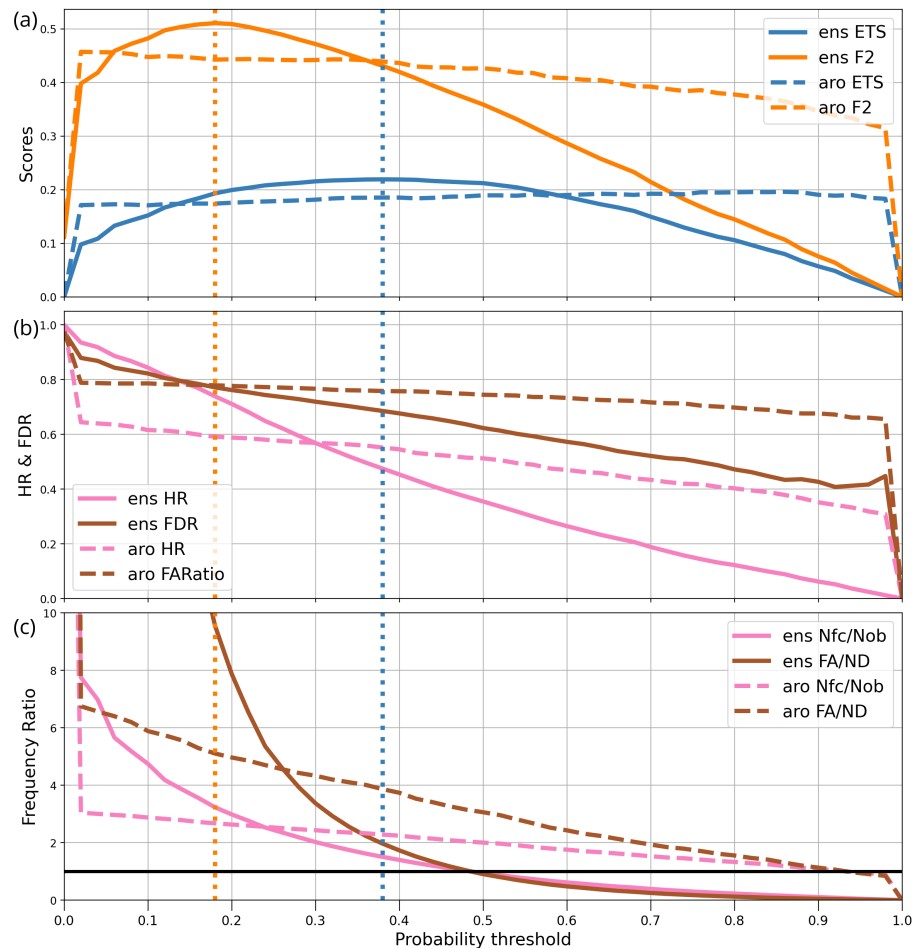

**Figure 4.** As in Fig.3, but for User H.

In conclusion, we have shown in this section that the proposed forecast optimization strategy works for two very different user types : the optimum scores are well defined, and they both indicate the superiority of decisions based on the ensemble forecast over the deterministic model. The performance of the optimal decision rules seems reasonable for weak precipitation (User L), but it seems lower for higher intensities despite using neighbourhood methods in the verification (User H). The following sections will now investigate the robustness of these conclusions with respect to space resolution, precipitation intensity, forecast range, season and accumulation time.

## 3.3 Sensitivity to the ensemble dressing operator

The impact of dressing on the ensemble post-processing was checked at the 30 mm intensity threshold with a 30 km neighbourhood and verification scale. The dressing kernel width was varied from zero to a large value (50%). Although the dressing





improves the hit rate, its net impact on the scores is very small, because most forecast errors are false alarms. The optimal dressing values are between 10 and 30%, and they only improve the CSI and F2 scores by 0.3% and 1%, respectively, compared to not dressing. Larger dressing widths were found to degrade the scores.

## 4   Sensitivity to the spatial neighbourhood operators

Spatial scale is an important aspect of the use and verification of precipitation forecasts (Ceresetti et al, 2012; Mittermaier,
2014; Ben Bouallègue and Theis, 2014; Buizza and Leutbecher, 2015; Schwartz and Sobash, 2017; Hess et al, 2018). The following sensitivity experiments investigate the impact on our results of changing the neighbourhood operators.

### 4.1   Sensitivity to the post-processing neighbourhood radius $R_f$

The optimization procedure has been rerun as in Sect. 3 with AROME-EPS forecasts, with fixed postprocessing parameters except for $R_f$, the forecast neighbourhood radius, which is varied between zero and 60 km.
The results for User L (Fig.5a) show that applying a neighbourhood to the forecasts only slightly improves the F2 and ETS scores with $R_f$ values of a few km. One may conclude that spatially filtering weak precipitation forecasts is unnecessary for User L.

The impact of $R_f$ on User H (Fig. 5b) is much larger. Both F2 and ETS increase with $R_f$ until a plateau is reached above 20 km, followed by a slow decrease above 40 km. Two reasons why $R_f$ is more beneficial for User H than User L can be
proposed. One is the conjecture by Schwartz and Sobash (2017) that the scale at which forecasts are upscaled should be consistent with the verification scale, which is the gridscale for User L and 30 km for User H. Another explanation is that 6-hourly precipitation errors above 30 mm are usually caused by thunderstorms with large location uncertainties. Most weak precipitation points are driven by more predictable synoptic fronts. Thus, precipitation location errors seem to affect User H more than User L.
Two practical conclusions can be drawn. One is that applying a forecast neighbourhood is beneficial when predicting high precipitation, and its radius $R_f$ should be set to values that are not too different from $R_o$. It has been checked (not shown) that this result holds for other precipitation thresholds and verification radii $R_o$. The other notable result is that $p_{\mathrm{opt}}$ is very sensitive to $R_f$: wider forecast neighbourhoods imply higher probability thresholds, because the max-neighbourhood operator widens the areas covered by the highest precipitation, which increases false alarms. As $R_f$ increases, $p_{\mathrm{opt}}$ increases as well
to compensate for this effect. Thus, when using neighbourhood post-processing, one needs to be careful to adjust $p_{\mathrm{opt}}$ as a function of the radius.

### 4.2   Sensitivity to the verification radius $R_o$

The previous study investigated the optimal forecast neighbourhood $R_f$ for users with fixed verification radius $R_o$. Here, we check how forecast scores change if the forecasts are verified at different resolutions. In particular, what happens if User H
uses forecasts at a higher resolution than $R_o = 30$ km ? To answer, a sensitivity study with respect to $R_o$ is now carried out as





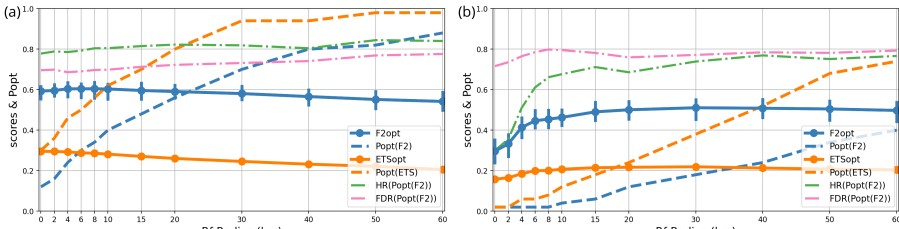

**Figure 5.** Sensitivity of the F2 and ETS scores to the forecast upscaling radius $R_f$ : respective optimum thresholds $p_{\mathrm{opt}}(\mathrm{F2})$, $p_{\mathrm{opt}}(\mathrm{ETS})$; optimum scores F2opt, ETSopt; error statistics HR, FDR at the F2 optimum. The statistics are shown for (a) User L and (b) User H. Vertical bars indicate 90% bootstrap confidence intervals on F2.

in the previous paragraph, except that we keep $R_f = R_o$ given that this choice of $R_f$ was shown to be optimal in the previous section. It has been checked that $R_f = R_o$ remains an optimal $R_f$ setting for other choices of $R_o$, but the corresponding plots are not presented for the sake of conciseness.

The results are shown in Fig. 6 using the 4 and 30 mm precipitation thresholds. In both cases, forecast scores F2 and ETS
strongly increase with verification scale. For instance, in terms of the F2 metric, 30 mm forecast accumulations are predicted 3.5 times better at 80 km scale than at the model grid scale, and both hit rate and false alarm statistics degrade quickly when going to finer resolutions. The hit rate drops sharply at scales below 20 km, which can be regarded as the finest resolution at which intense precipitation can be skilfully predicted by the AROME model.

At lower precipitation intensities, the dependency to resolution is similar, but weaker : at the 4 mm threshold, the ETS metric
is only 1.6 times better at 80 km scale than at model grid scale. In other words, precipitation is more predictable at larger scales than smaller ones, which is consistent with the findings of Buizza and Leutbecher (2015) that the predictability of atmospheric variables increases with the horizontal scale considered. This scale dependency is most pronounced for the highest intensity precipitation events, which can be attributed to their larger location errors.

Two practical conclusions can be drawn. First, if one wants forecasts to meet some minimum accuracy requirements, one
should not use model output at too fine resolutions, because the finest details tend to have large forecast errors. Second, because $p_{\mathrm{opt}}$ has little sensitivity to $R_o$ above 15 km, it can be considered to be independent from the scale at which forecasts are used, once the finest unpredictable details have been filtered out.

## 5 Sensitivity to precipitation intensity

In this section, the extrapolation of the previous results to the highest available precipitation intensities is investigated. The
postprocessing and verification radii are kept to $R_f = R_o = 30$ km, which has been shown to be a satisfactory setting for a wide range of intensities. The 6-hourly precipitation threshold is varied between 1 and 80 mm. Above 60 mm or so, forecast information is valuable because catastrophic consequences begin to occur (with a return period of several years, see Fig.1b), but the number of independent cases available to compute statistics decreases rapidly. The intent is to assess if the relatively





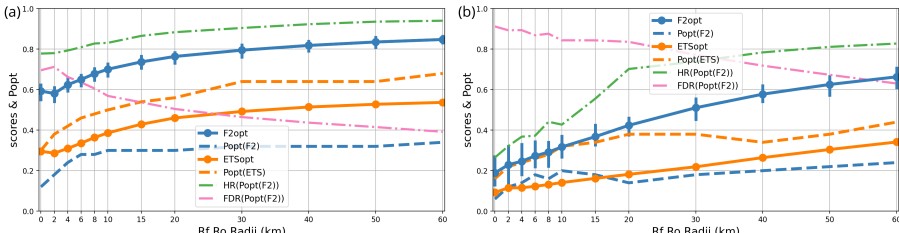

**Figure 6.** As in Fig.5, as a function of $R_o = R_f$, the verification and post-processing radii which are kept equal to each other.

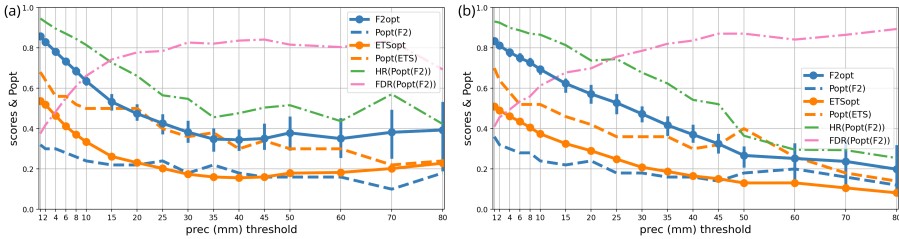

**Figure 7.** As in Fig.5, as a function of the precipitation intensity threshold, for User H over 2 periods: (a) September to November 2022 and (b) May to July 2023.

robust statistics shown in previous sections ($p_{\text{opt}}$ in particular) can be extrapolated to higher intensities that occur too rarely for a direct statistical treatment. As our focus is on heavy precipitation, and it has been shown that forecast accuracy is very low at verification scales below 20 km, we only present results for User H, at a verification scale of $R_o = 30$ km.

Figure 7 shows the precipitation forecast performance as a function of intensity. The statistics have been gathered over two independent 3 month periods, as opposed to one month in Sect. 3, in order to improve the sampling of higher intensity events. Even so, there is substantial sampling uncertainty at 60 mm and above, as illustrated by the 5%–95% bootstrap confidence intervals displayed on the F2 curves.

During both periods there is a steep degradation of the scores as the intensity threshold increases. Light accumulations are much more predictable than heavy ones that exhibit more than 80% of false alarms and less than 50% detection rates at the highest thresholds. A user who requires better forecast performance than these values may conclude that numerical predictions are not suitable, and that forecast decisions should instead be based on other data sources like nowcasts or observations.

Both tested periods have fairly similar statistics, and the score variations are simultaneously caused by detections and false alarms. Interestingly, the autumn scores are nearly constant over 30 mm, but spring scores keep decreasing at all thresholds. A possible explanation is that heavy precipitation in autumn is more likely to be caused by orographically forced convection in the Mediterranean area. These events are driven by slowly evolving, highly coherent weather patterns (Amengual et al , 2021; Davolio et al, 2013; Nuissier et al, 2016; Caumont et al, 2021), with comparatively high predictability.



The $p_{\mathrm{opt}}$ statistic strongly depends on intensity, but its variations are smooth: for F2, it is close to 0.3 at low intensities, and it decreases to about 0.15 at 80 mm (depending on the period). Above 30 mm, $p_{\mathrm{opt}}$ values are nearly constant and vary by less than 5%, which is inside the tolerance intervals identified in Sect. 3. Thus, it seems that issuing precipitation forecasts using the 0.85-quantile of the ensemble forecast is a nearly optimal strategy for all precipitations above 30 mm, if the aim is to maximize the F2 score at a minimum scale of 30 km.

## 6 Sensitivity to the time dimension

This section investigates the robustness of the previous results with respect to three aspects of the forecasts : forecast range, season, and accumulation length. The diurnal cycle will be studied through the forecast range dependency, since all forecasts considered start from the same time of day.

### 6.1 Sensitivity to forecast range and diurnal cycle

Fig. 8 shows the forecast statistics of 6-hourly accumulations for User H, averaged over 14 months (July 2022 to September 2023), and stratified against the AROME-EPS forecast range. More forecast ranges have been taken into account in this plot than in previous sections, in order to adequately sample the time evolution.

The shortest range considered is 9 h (accumulated precipitation at ranges from 3 to 9 h), because the AROME-EPS operational production typically takes a little over 2 h to run in real time, so ranges smaller than 2 h are useless. Second, the 415 first timesteps have poor error statistics because of diabatic spinup effects in the AROME model. Spinup occurs because of physical inconsistencies in data assimilation and in initial ensemble perturbations (e.g., Bouttier et al, 2016; Brousseau et al, 2011; Auger et al, 2015). Although the highest operational forecast range is 51 h, to save computations our study is limited to a range of 36 h. The range interval used for this study (3 to 36 h) is long enough to sample the whole diurnal cycle.

Fig. 8 shows that the scores and statistics are nearly independent from range. There is a small decrease of less than 10% in 420 F2, ETS from 9 to 36 h (12 to 15 local solar time), hit rate and $p_{\mathrm{opt}}$. A hint of diurnal cycle is visible. The lower predictability at ranges 15 to 18 is probably due to diurnal development of convection during the warm season. Nevertheless, the diurnal cycle is too small to have practical significance.

Published verifications of ensemble predictions have usually shown that their scores decrease over ranges of several weeks due to chaotic error growth during the forecasts (Buizza and Leutbecher, 2015). During the first forecast day, the relative 425 decrease of the scores tends to be small (Bouttier et al, 2016). Thus, over the ranges studied here, chaotic forecast error growth is hardly visible because it occurs over longer timescales. At short ranges, the quality of the AROME-EPS initial conditions is driven by the application of relatively large initial ensemble perturbations (Bouttier et al, 2016), which lead to a comparatively slower error growth than in nowcasting-oriented forecast systems (Auger et al, 2015), where the emphasis is on consistency of the initial conditions with the latest observations. Combining nowcasting information with AROME-EPS could probably 430 improve the short-range forecasts, as suggested by Nipen et al (2011).





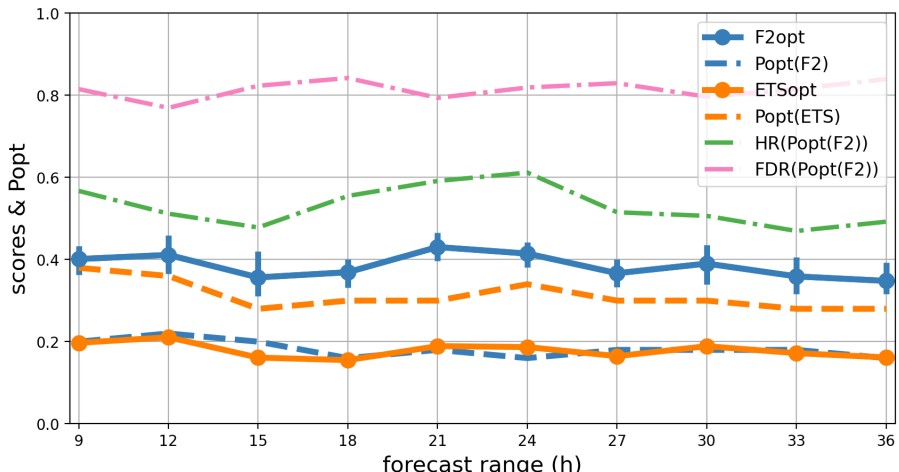

**Figure 8.** As in Fig.5, as a function of forecast range and on average over one year, for User H.

In conclusion, one can consider for practical purposes that the scores and optimal thresholds are nearly independent from the forecast range and time of day.

## 6.2 Seasonal dependency

The error statistics are now shown as a function of the forecast starting date over 15 months in Fig.9; a 3-month running average has been applied. Again, the variations are small, apart from a kink around February and March 2023. It was an exceptionally dry period, so this feature does not seem significant. The general shape of the curves seems driven by the strong natural variability that occurred at the monthly timescale during the period: the weather was dry except for a few rainy spells during Aug–Nov 2022, May–Jun 2023 and Sep 2023. The conclusion is that, if there is a seasonal cycle in the scores and in $p_{\mathrm{opt}}$, it cannot be clearly identified by this study. A longer dataset covering several years would be needed in order to average out the natural variability that occurs from month to month, for identification of seasonal dependency in the scores and the associated $p_{\mathrm{opt}}$ thresholds.

## 6.3 Sensitivity to accumulation timescale

Conceptual models of turbulent flows suggest that smaller-scale features have shorter life span and lower predictability that larger-scale ones; this view is supported by atmospheric predictability studies (e.g., Buizza and Leutbecher 2015). Hence, one can suspect that the precipitation forecast scores and the associated decision thresholds $p_{\mathrm{opt}}$ depend on the length of the accumulation windows, which act as a low-pass time filter. To check this hypothesis, the same 24 h interval of forecast ranges from 3 to 27 h has been split into six different partitions: $1 \times 24$ h, $2 \times 12$ h, $4 \times 6$ h, $8 \times 3$ h, $12 \times 2$ h and $24 \times 1$ h. The forecast error statistics have been computed on accumulations over these intervals.



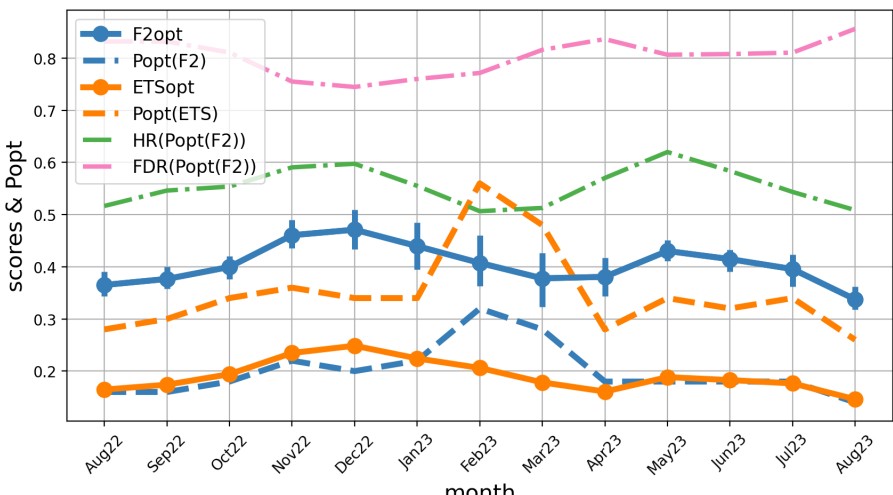

**Figure 9.** As in Fig.5, as a function of forecast starting month, for User H.

| accumulation time (h) | 1 | 2 | 3 | 6 | 12 | 24 |
|---|---|---|---|---|---|---|
| precipitation with 5 year return period (mm) | 30 | 40 | 50 | 60 | 65 | 70 |

**Table 1.** Precipitation thresholds with a 5 year return period used in Fig.10.

For each accumulation period, the intensity thresholds for computing scores were adjusted in order to represent similar levels
of severity, defined as the spatial median of the 5 year return period of precipitation in the SHYREG climatology (Fig. 1b). The
thresholds used are shown in Table 1.

The scores and $p_{opt}$ statistics have been computed over two independent 3 month periods, chosen for their relative abundance
of precipitation: September to November 2022, and May to July 2022. The first one was characterized by a strong synoptic
forcing with intense orographically forced precipitation events. The second one had mostly weakly forced thunderstorms.

The dependency of the statistics with respect to accumulation time is shown in Fig. 10. During the Sep–Nov 2022 period,
longer accumulations are much more predictable than shorter ones (the F2 and ETS scores are higher, the hit rate is higher and
the probability of false alarm is lower). Accordingly, $p_{opt}$ is nearly twice as high for 24 h accumulations as for 1 h ones.

This result is important for hydrological applications, because it indicates that flood warnings driven by such an ensemble
prediction system should use probability thresholds $p_{opt}$ that depend on the timescale that is most relevant to each river system.
Our results show that floods that depend on timescales longer than 6 h can have forecast error statistics that are quite different
from shorter ones. Heavy rain warnings for accumulations over 1 to 3 h need to be triggered at very low probability levels (10%



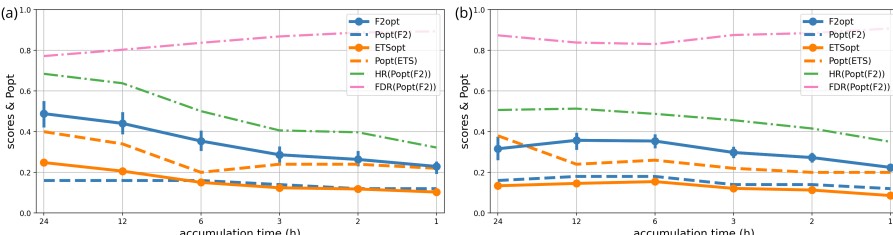

**Figure 10.** As in Fig.5, as a function of accumulation time, for User H only, over (a) Sept–Nov 2022 and (b) May–Jul 2023. Precipitation thresholds were adjusted to match the 5 year return period (see text and Table 1)

or less) because their hit rates are quite low. Forecasting of such short-timescale events should probably rely on nowcasting products and/or be used with some time tolerance.

During the May–Jul 2023 period, the statistics are nearly independent from the timescale and not better than the shortest
(i.e., the worst) timescales of the other period. A possible explanation is that this period is dominated by intense but non-stationary rainy systems such as thunderstorms : their accumulations over longer periods are produced by multiple short-lived, unpredictable convective systems, unlike the persistent rains from long-lived coherent weather systems.

The conclusion is that longer accumulations are generally more predictable than shorter ones, and that this information could be used to optimize warning decisions, particularly regarding flood risks. Unfortunately, this effect seems to depend on
the period considered, so that a more elaborate model of this dependency (possibly with predictors of weather type) should be introduced into the $p_{\mathrm{opt}}$ computations before this information can be confidently used in practice.

## 7  Case studies

The previous results suggest that automatic precipitation exceedance warnings could be obtained from the optimal ensemble postprocessing parameter values using probability threshold $p_{\mathrm{opt}}$. Human forecasters are usually responsible for issuing severe
weather warnings on the basis of their expert knowledge. Hence, they need to understand the meteorological context of the numerically predicted information, which can be visualized from weather maps. Inspecting ensemble members can be tedious: there is active research on designing efficient graphical products to present ensemble forecast information (Demuth et al, 2020). Here, we focus on the problem of graphically representing the information that a high precipitation warning should be issued because the ensemble forecast probability is larger than $p_{\mathrm{opt}}$. Probability exceedance maps are not ideal because they do not
show a physical parameter (the probability value) and they are defined with respect to a fixed threshold.

Our results have shown that, for high precipitations, $p_{\mathrm{opt}}$ is a slowly varying function of precipitation intensity, so we propose to summarize the ensemble prediction by maps of (1-$p_{\mathrm{opt}}$)-quantiles of the post-processed ensemble. These maps show precipitation intensities that have probability $p_{\mathrm{opt}}$ of occurring. We will call them "optimal quantiles" for the user for whom $p_{\mathrm{opt}}$ has been optimized. At each grid point, the optimal quantile is the precipitation value that is most likely to minimize
user losses. In our framework, User L should, on average, rely on maps of $q_L$-quantiles with $q_L = 1 - p_{\mathrm{opt}}(\mathrm{ETS}) = 0.7$, with



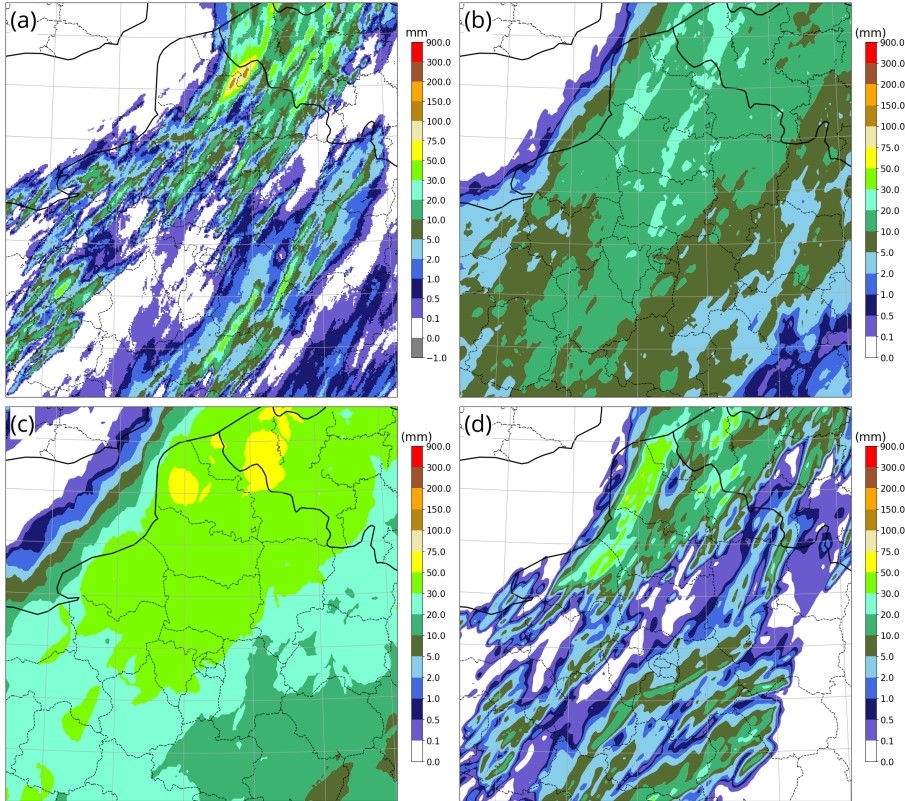

**Figure 11.** Maps of 6-hourly accumulated precipitation on June 20th, 2023, between 12 and 18utc, over northern France. (a) Truth provided by the ANTILOPE analysis, (b) $q_L$-quantile of the AROME-EPS ensemble 21 h range forecast (0.7-quantile with 5 km neighbourhood), (c) $q_H$-quantile of the same forecast (82% percentile with 30 km neighbourhood), (d) AROME-France deterministic forecast.

a small neighbourhood (set to $R_f = 5$ km), which is appropriate for gridpoint predictions where non-detections carry the same costs as false alarms. The optimal quantile for User H is the $q_H$-quantile with $q_H = 1 - p_{\text{opt}}(\text{F2}) = 0.82$, with a neighbourhood of $R_f$=30 km, which is appropriate for severe weather warnings at this spatial scale, incurring approximately 4 times more false alarms than non-detections.

These maps are now presented on several test cases. The first one, illustrated in Fig.11, involved a highly localized flood event at the border between France and Belgium. Observed accumulations exceeding 100 mm were produced by a line of mobile thunderstorms embedded in a synoptically driven cold front. The large scale features of the front were highly predictable, but the small-scale details were produced by small-scale convection with large uncertainties in location and timing, as reflected in the ensemble spread and in the forecast errors. Although most ensemble members predicted significant precipitation (a 5 year

return period was reached by the deterministic AROME forecast), they were all weaker than observed.

The spatial ensemble spread of the predicted thunderstorms was much larger than their size, so that most quantile maps, such as the $q_L$-quantile shown in Fig 11b, did not show precipitation above 30 mm. The $q_H$-quantile in Fig 11c shows much




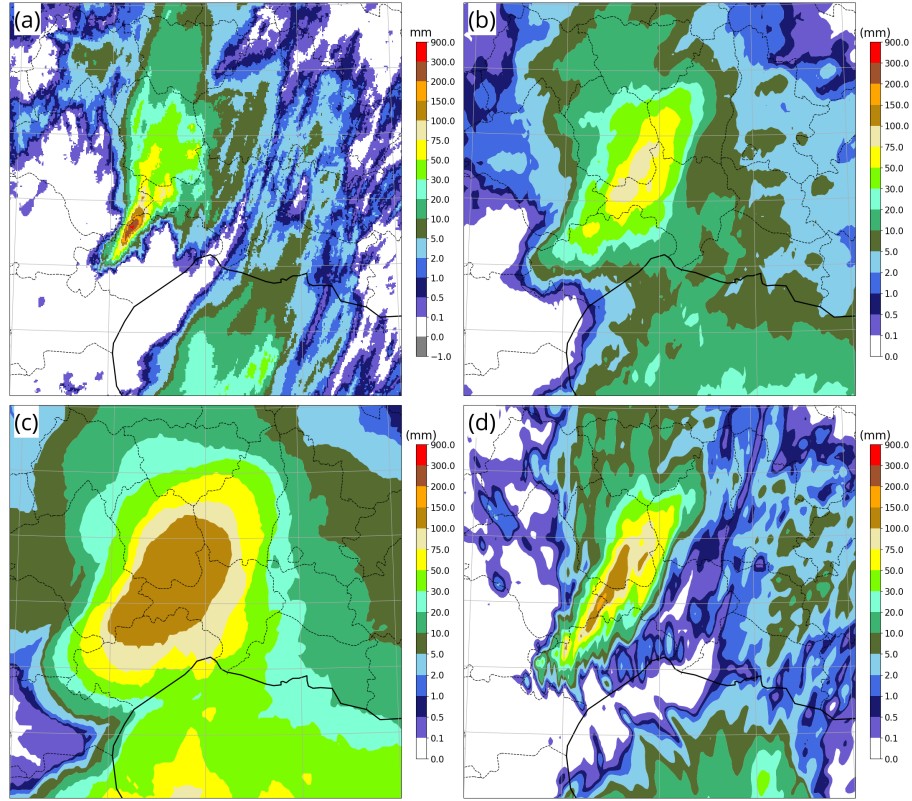

**Figure 12.** As in Fig.11, on Sept. 16th, 2023, between 00 and 06utc, over southeastern France.

more realistic intensities, as well as a more informative delineation of the areas where severe precipitation is likely to occur. For instance, the $q_H$-quantile areas with values above 50 mm correctly framed the observed high precipitation event (although

the peak observed intensity was much higher than predicted), thanks to the combination of the ensemble spread with the neighbourhood postprocessing. A high precipitation warning based on $q_H$-quantile values above 50 mm would arguably result in many false alarms because of the limited sharpness of the ensemble forecast, but it nevertheless would still contain better information about the areas at risk than the deterministic forecast (Fig 11d). The latter predicted similar values, but much further away (by more than 70 km) from the observed event.

The $q_L$- and $q_H$-quantile forecasts are quite different from each other, which demonstrates that in the presence of high forecast uncertainties, the optimal forecast can strongly depends on the user for whom the forecast is optimized.

    The second case, presented in Fig.12, is typical of flooding events that frequently occur in autumn around the northwestern Mediterranean sea. In such cases, synoptically forced warm and moist low-level onshore winds trigger quasi-stationary convective systems above or near hills. Here, over 300 mm of rain were observed over 6 h in a very localized area (Fig.12a) of

the Cévennes mountains (located where the highest values of Fig. 1b occur). As in the previous case, the $q_L$-quantile severely underestimated and misplaced the highest accumulations, while being a rather accurate forecast in weaker precipitation areas.



The $q_H$-quantile successfully encompassed the observed high precipitation event with more accurate values. The deterministic forecast was more precise than the $q_H$-quantile in terms of false alarm avoidance and maximum values, but its peak was located too far north, missing the observed peak by about 20 km .

It is not clear whether the best overall forecast was provided by the $q_H$-quantile or by the deterministic forecast. The $q_H$-quantile detected the whole event at the price of a less precise framing of the area at risk (leading to many false alarms), and of an underestimation of maximum intensities: answering is a matter of preference between detection, false alarms, and peak intensity. The likelihood of high precipitation in this region has a climatological NW–SE gradient (see Fig. 1b), so that the $q_H$-quantile forecast could probably be improved (with less blurring) by using anisotropic neighbourhoods in this area, to express

the fact that location errors are more likely to occur along the SW–NE than along the NW–SE directions.

The third case, presented in Fig.13, was an unstructured afternoon thunderstorm case in a convectively unstable airmass. Unlike the previous cases, it was not caused by significant forcing from the synoptic cases or from orography. Many small, quasi-stationary convective cells developed chaotically, some of them leading to large accumulations at highly unpredictable locations. The AROME deterministic forecast (Fig.13d) accurately predicted the precipitation maximum and the overall pre-

cipitating region. The texture of the field looked more realistic, with a good prediction of the size of precipitating cells, but their individual locations were generally wrong. There was a general northward shift of up to 100 km. Some large areas that were predicted to be dry actually experienced thunderstorms. The $q_L$-quantile (Fig. 13b) correctly smoothed out most random small scale detail. It was a more satisfactory prediction of rain occurrence than the deterministic forecast, but it severely underpredicted the accumulations under the thunderstorms. Conversely, the $q_H$-quantile (Fig. 13c) overpredicted the light precipitation

area, but it correctly delineated the envelope of high rain area, with a better indication of the highest expected accumulation than both AROME and the $q_L$-quantile.

In conclusion regarding the case studies, we have demonstrated than our technique for computing the $q_L$- and $q_H$-quantiles leads to quite different forecast maps. They both eliminate unpredictable small-scale detail from the member forecasts, as illustrated by the comparison with the deterministic AROME output. Maps of the $q_L$ and $q_H$-quantiles are very different

from each other because they cater for different needs : the $q_L$-quantile summarizes information about rain occurrence and light accumulations, whereas the $q_H$-quantile suggests a worst case scenario in terms of extent and amplitude of the heaviest accumulations. The added value of the $q_L$- and $q_H$-quantile products with respect to the raw deterministic forecast is not always clear: it may not be obvious if the weather situation is highly predictable (e.g. the location of moderate precipitation in Fig.12), or if all members exhibit the same kind of errors. In the latter situation, similar errors will be found in the post-processed

products because our algorithm is not designed to correct for systematic forecast errors.

Visual inspection of other high precipitation cases has suggested that the $q_L$- and $q_H$-quantiles do not always convincingly improve over the deterministic forecast, but they do not significantly degrade it either, which suggests that optimal quantiles are a safe way of summarizing ensemble forecasts for users with well defined objectives.



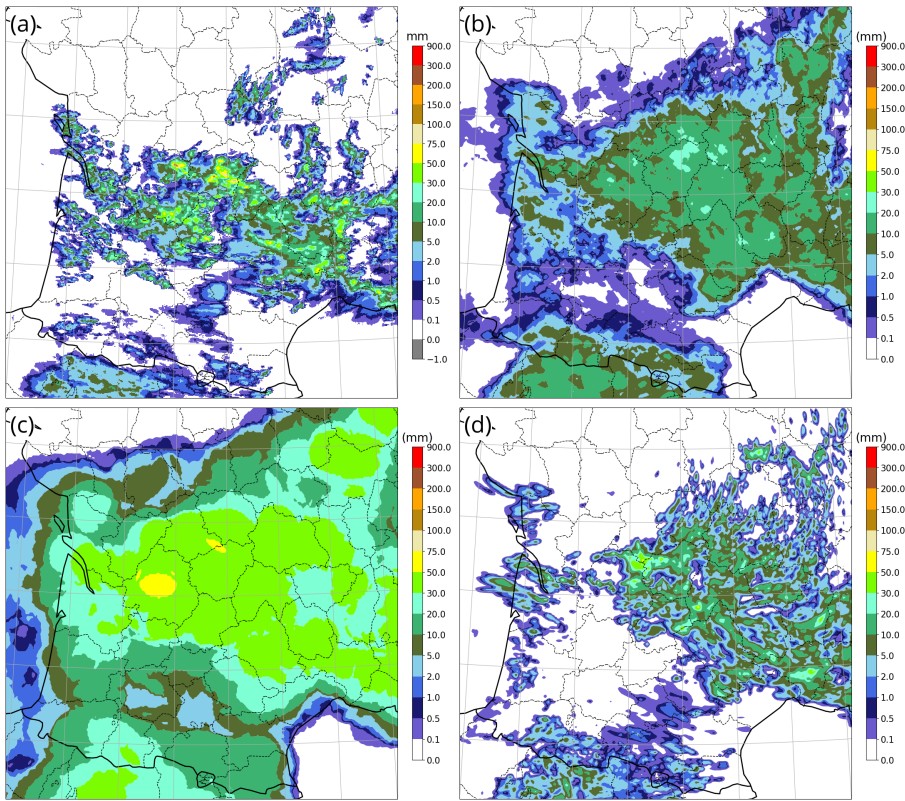

**Figure 13.** As in Fig.11, on May. 22th, 2023, between 12 and 18utc, over southwestern France.

# 8   Conclusions

A data based ensemble postprocessing method has been presented. It is a generalization of the proposals of Ben Bouallègue
and Theis (2014) and Schwartz and Sobash (2017). It takes into account user requirements in terms of target rain intensities
and tolerance to false alarms vs event non-detection. The aim is to summarize ensemble forecasts for efficient time-critical
interpretation and as an input to decision making such as high precipitation warnings.

The method is a statistical learning based on forecasts archived over a few months. Its output is a set of optimal deci-
sion thresholds expressed as an ensemble probability threshold $p_{opt}$ (or, equivalently, an optimal quantile level). The post-
processing combines a probability dressing step, a maximum spatial neighbourhood operator (to represent the effect of location
errors), and a probability threshold optimization defined with respect to a user-dependent loss function. The dressing step al-
lows a fair comparison between ensemble and deterministic forecasts. The loss functions model the requirements of two kinds
of users based on the ETS and F2 scores. The properties of F2 make it suitable for scoring high-impact weather events with a
higher tolerance to false alarms than to event misses.



Using impact experiments, the sensitivity of the $p_{\mathrm{opt}}$ optimization to several aspects of the post-processing has been examined to ensure that the procedure is stable, and that its sensitivity to the implementation settings is well understood. The value of $p_{\mathrm{opt}}$ turned out to be sensitive to the choice of loss function, precipitation intensity and neighbourhood radius, but not much to forecast range, season and precipitation accumulation time. The $p_{\mathrm{opt}}$ variation with precipitation intensity is smooth, so that it can be extrapolated from the highest intensities allowed by the training sample, to the extreme precipitation forecasts produced by a convection permitting numerical ensemble. Case studies in high precipitation events support this extrapolation hypothesis to the extent that, for each targeted user, $p_{\mathrm{opt}}$-based optimal quantile forecast maps seem to better summarize useful ensemble information than a deterministic control forecast.

Objective scores of $p_{\mathrm{opt}}$-based forecasts are significantly better with an ensemble forecast than a deterministic control forecast. The inclusion of the neighbourhood operator improves forecasts of relatively high precipitation events, but its impact on forecasts of low precipitation is small. The set of forecast ranges considered in our study is too narrow to document the chaotic loss of predictability in medium range (2 to 15 days) forecasts. This loss appears to be negligible from 3 to 36 h.

Three kinds of relative "predictability horizons" for precipitation can be identified: intensity, time and space resolution. After $p_{\mathrm{opt}}$ optimization, precipitation forecast skill generally decreases with intensity, with the possible exception of orographically forced events. It increases with accumulation time, although this dependency appears to be situation-dependent. It strongly increases with verification scale. These results confirm the physical intuition that larger scales (in space and time) are more predictable than smaller ones. The higher precipitations are less predictable, presumably because they tend to be produced by relatively small scale weather systems.

The score variations are clearly related to hit rates and probabilities of false alarm, so that one can conclude that users expecting forecasts to meet predefined performance requirements should not ask for too fine-scale forecasts. In particular, high precipitation forecast scores drop quickly at scales smaller than 10 to 30 km. Although there is some skill in the prediction of high precipitation, users wishing to base their decisions on confident high precipitation forecasts (such as emergency services with constrained resources), or forecasters who wish to avoid losing credibility by issuing false alarms too often, may conclude that they should not base their forecast decisions on numerical weather prediction and they might prefer to rely on more accurate data sources such as nowcasting.

The post-processing method proposed here can be regarded as an ensemble calibration, since our $p_{\mathrm{opt}}$ optimization procedure will compensate for some forecast biases. Standard ensemble calibration methods such as published by Gneiting et al (2005), Scheuerer (2014) or Flowerdew (2014) have usually been designed to optimize generic aspects of the forecasts. The originality of our work is to directly optimize the post-processing in terms of user specific loss functions which guarantees that they get the best possible information. As our sensitivity and case studies have demonstrated, when forecast uncertainties are significant, probabilistic forecast products strongly depend on the preferences of the users for which they are designed.

Fundel et al (2019) and Demuth et al (2020) pointed out that the variety of user needs is a challenge for communicating forecast uncertainty information. The method outlined in our work proposes to help with this problem by summarizing forecasts using maps of optimal quantiles, which are relatively quick and easy to interpret : they are expressed in physical units and their relationship with the member fields can be intuitively retraced based on $p_{\mathrm{opt}}$ and the neighbourhood radius. These maps can be





generated from limited historical archives. The information they convey is objectively better on average than the corresponding deterministic forecast, and they have the potential to help identify regions with high precipitation risks in various types of weather situations.

Two applications of this work can be envisioned. First, the $p_{\mathrm{opt}}$ information, in particular as optimal quantile maps, could help human forecasters to quickly access ensemble forecast information in terms of the geographical extent and intensity of high precipitation events. Second, this information could be included into automated point weather forecast applications to ensure that they convey the risks of high precipitation, even if the forecast accuracy of our algorithm is still probably much lower than what experienced humans can achieve in terms of severe precipitation warnings.

The proposed technique could be improved in several ways, which is left for future work. For instance, more members could be used by combining multimodel ensembles with lagging. Nowcasting information could be included at short range. The neighbourhood operator could be made anisotropic and location dependent in order to account for orographic effects. Neighbourhood operators that depend on ensemble spread have been tested by Blake et al (2018) with encouraging results. More fundamentally, $p_{\mathrm{opt}}$ could be computed by a regression in a more general statistical model, for instance a convolutional artificial neural network that would include geographical and seasonal dependency and indicators of weather type as predictors, besides the precipitation fields used in this study.

The loss function could also be improved to include other measures of forecast quality than the numbers of non detections and false alarms. For instance, it may be helpful to minimize inconsistency between successive forecasts (Demuth et al, 2020), as well as repeated false alarms that can lead to user loss of confidence in the forecasts.

*Code and data availability.* The figures of this article can be regenerated from the data provided at a public repository under doi:10.5281/zenodo.10420739 with the exception of Fig.1b which uses proprietary data that can be requested from the INRAE institute.

*Author contributions.* Conceptualisation and methodology: FB and HM; investigation, software, interpretation, visualisation and writing: FB. All authors have read and agreed to the published version of the manuscript.

*Competing interests.* The contact author has declared that none of the authors has any competing interests.

*Acknowledgements.* This work was funded by the French government through Météo France and CNRS.



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
