# Peer review of "Probabilistic short-range forecasts of high precipitation events: optimal decision thresholds and predictability limits"

_EGUsphere, 2023_

## Referee Comment (RC1)

**EGU-2023-3111**

**Probabilistic short-range forecasts of high precipitation events: optimal decision thresholds and predictability limits**

*Bouttier and Marchal*

*Overall:*

The paper presents an interesting approach to deriving optimal decision thresholds for probabilistic ensemble forecasts for two users with different risk appetites and requirements. It provides a means of moving the decision-making process upstream, away from the user, back into the meteorological sphere/space, where the expertise, in terms of interpretation is more likely to exist.

Whilst some in the community may be horrified at collapsing a probability forecast down to a deterministic one, it is what the user does, and as the authors point out, having a sense of where the ensemble adds value over the deterministic is important because it is not a given that the ensemble will add value. In the end most users will either have to choose to do something or nothing. This is a binary decision point.

The lack of skill is clear and potentially disheartening. However, the study provides some tangible evidence of the current state-of-the-art capability in our NWP systems and how to maximise the value they can provide.

I recommend the paper is accepted with some minor revisions.

*Specific comments:*

Please check the figure numbering and text where the figures are referred to. I did think there are a number of places where things are either not referred to or not referring to the correct figure.

L113-115: I rather like how you have worded this and would like to see you use something similar in the abstract. It would strengthen the abstract in my view.

L179: Perhaps more a comment than anything else, but I would be very wary of applying a neighbourhood verification method on top of using a neighbourhood post-processing approach first.

L186 and L191: the 4- and 30-mm thresholds appear somewhat at random here. Is there any rationale for these values?

L219: I think you meant that it gives 4 times more weight to c than to b?

L220: I would replace non-detections with missed events

L260: I feel this "only" is misplaced given that 0.12 < 0.3

L271: missing bracket

L274: I think you mean Fig 3 here

L295: great illustration of the value of the FDR over POFD

L317: I am not that familiar with the F2 score but we know that the ETS can be optimised by over-forecasting. It looks like the F2 can as well. Over-forecasting is by far the most useful mechanism for reducing missed events (of course at the expense of the false alarms!)

L320: This is probably an important result and one worthy of elevating more in the abstract perhaps? For user H the ensemble probably has more value, even if the scores are lower.

L332: This would be consistent with the notion that whilst spread is necessary, too much spread is not good.

Figs 5, 6 and 7 are very small. The final version should have bigger panels. These figures contain some important findings.

L343: What Fig 5b shows really well is how the use of some sort of neighbourhood improves the score.

L346-349: You'd hope that the ensemble (especially) can capture some of that spatial uncertainty. Not all heavy rainfall events are convective. Perhaps it would be better to say, that if 30 mm/6h are convection-driven, these come with large spatial uncertainties. To say that location errors affect user H more than L, is perhaps a little disingenuous because user H is a subset of user L. I strongly suspect that if user L was hit by the that 30 mm/6h they would still be affected by it!

L365-368: This is the double penalty effect in action and why we use upscaling.

L383: it might be useful to see what N is here for each of the thresholds

L417: suggest longest instead of highest

L435: What it does suggest is that prolonged periods without rain/events can be detrimental to aggregated statistics. It is therefore not the length of time that is important but the number of events in that time window.

Fig 10. I am not entirely sure I like the highest totals to the left. It is somewhat counterintuitive.

L501: I guess what this example tells me is that the basic characteristics of the forecast (ensemble or deterministic) are still very much fundamental to the success of applying a user-relevant decision threshold. The outcome will only be as good as the forecast itself.

L516: This is a crucial point. Many countries are constrained in not issuing warning areas which are too large. Therefore, having specific and tight warning areas is highly desirable for many met services.

L528-531: Perhaps one of the good things about this way of post-processing the forecasts is that it does provide this alternative "views" of the same forecast.

---

## Referee Comment (RC2)

**Review to EGUsphere-2023-3111 "Probabilistic short-range forecasts of high precipitation events: optimal decision thresholds and predictability limits".**

In their manuscript, the authors present a new data based ensemble post-processing method to summarize ensemble forecasts as an input to e.g. high precipitation warnings. This new approach takes into account user requirements in terms of target rain intensities and tolerance to false alarms and event non-detection. The method is based on a statistical learning of forecasts archived over a few months. The output is a set of optimal decision thresholds expressed as an ensemble probability threshold or an optimal quantile level. To examine the sensitivity of the procedure, several impact experiments have been made and furthermore 3 real case studies have been presented.
The authors found out that ensemble predictions objectively outperform the corresponding deterministic forecasts at low precipitation intensities. Precipitation forecast skill generally decreases with intensity (expect for orographically enhanced events), while it increases with accumulation time, and verification scale. The post-processing method proposed in this manuscript can also be regarded as a kind of ensemble calibration since forecast biases are compensated.

The paper presents an interesting approach which could be helpful for human forecasters as a kind of supporting tool in case of severe weather but could also be included into automated forecast applications. The structure of the manuscript is clear and conclusive and it is written in excellent English supported by well- arranged figures (some of them should be enlarged) and tables.
The article is addressing topical methods in probabilistic weather forecasts/warnings and hence fits very well into the scope of the journal.
However there are some things that should be addressed before final publication.
I recommend the paper is accepted with some minor revisions.

General: There are a lot of blanks in the text (before .,:) which should not be there. Also some inconsistencies (e.g. abbreviations, figure references, numbering, etc.) are present. Check the whole manuscript with respect to that.

1. Page 2, L38: significance or skill measures. Typo.
2. Page 3, L82: Reference to Wikipedia. Can you add a more scientific reference to this statement.
3. Page 3, L90: "prediction scales of the order of 30km, which is the resolution at which weather warnings are often issued in European countries. Add some reference for this statement.
4. Page 4, L121: The following paper will be structured as follows. Missing word.
5. Page 4, L122: Section 6 is described twice in this sentence.
6. Page 5, L138: What do you mean by "upper boundary conditions"? upper air? Describe it.
7. Page 7, L169: blank is used before : This is in the whole text, sometimes you have a blank before :, sometimes not. Be consistent.
8. Page 7, L186: is there a reference why you use 4mm and 30mm as threshold? Add reference or description.
9. Page 8, L198: produce. Typo.
10. Page 9, L227: A word is missing in this sentence (e.g. of these terms).
11. Page 10, L267: "should not be interpreted as a …" Missing word.
12. Page 10, L271: Missing bracket.
13. Page 10, L275: You mean Figure 3a – wrong reference to figure.

14. Page 15 Figure 5: too small figure – should be enlarged.
15. Page 16 Figure 6 and 7: too small figures – should be enlarged.
16. Page 16, L396: This is strongly depending on the region, since seasonal precipitation characteristics are completely different between the northen parts of France and the Mediterranean region. This should be stressed more.
17. Page 17, L 413: This sentence is confusing – add 6h-accumulated precipitation…
18. Page 17, L420: Why 12 to 15 local solar time? I do not understand, I thought you are using 21 UTC runs.
19. Page 18, section 6.2 As mentioned before, this seasonal dependency could be very different from region to region – a statement about this would be fine.
20. Page 19, L461: But this will cause a lot of false alarms.
21. Page 20 Figure 10: too small figure – should be enlarged.
22. Page 20, L475: and forecasters on duty do normally not have the time to look on each members separately.
23. Figures 11, 12, 13: UTC should be written in capital letters (in the complete manuscript).
24. Page 20, L504: On the other hand you have an overestimation in the central parts of the domain in Figure 11 in qH quantile plot compared to the deterministic plot and the observations.
25. Page 21, L506: can strongly depend. Remove the s in depends.
26. Page 23, L515: It would be interesting to have a "classical" ensemble median or probability map in comparison to qL and qH quantiles plots for this case.
27. Page 23, L532: that in instead of than.

---

## Author Response (AR1)

**Response to first round of reviews on manuscript https://doi.org/10.5194/egusphere-2023-3111 'Probabilistic short-range forecasts of high precipitation events: optimal decision thresholds and predictability limits' by F. Bouttier and H. Marchal**

*Response to reviewer RC1: 'Comment on egusphere-2023-3111', Anonymous Referee #1, posted 01 Mar 2024, https://doi.org/10.5194/egusphere-2023-3111-RC1 (author's response below is typeset in italics)*

*We thank the reviewer for the thorough and very helpful comments.*

**EGU-2023-3111**
**Probabilistic short-range forecasts of high precipitation events: optimal decision thresholds and predictability limits**
*Bouttier and Marchal*
**Overall:**

The paper presents an interesting approach to deriving optimal decision thresholds for probabilistic ensemble forecasts for two users with different risk appetites and requirements. It provides a means of moving the decision-making process upstream, away from the user, back into the meteorological sphere/space, where the expertise, in terms of interpretation is more likely to exist.

Whilst some in the community may be horrified at collapsing a probability forecast down to a deterministic one, it is what the user does, and as the authors point out, having a sense of where the ensemble adds value over the deterministic is important because it is not a given that the ensemble will add value. In the end most users will either have to choose to do something or nothing. This is a binary decision point.

The lack of skill is clear and potentially disheartening. However, the study provides some tangible evidence of the current state-of-the-art capability in our NWP systems and how to maximise the value they can provide.

I recommend the paper is accepted with some minor revision.

*We agree with the recommendation and respond point by point to the specific comments below.*

**Specific comments:**

Please check the figure numbering and text where the figures are referred to. I did think there are a number of places where things are either not referred to or not referring to the correct figure.

*Sorry that some figure references were wrong, they have been corrected.*

L113-115: I rather like how you have worded this and would like to see you use something similar in the abstract. It would strengthen the abstract in my view.

*A corresponding sentence has been added to the abstract: "We propose an ensemble-based deterministic forecasting procedure that can be optimized with respect to spatial scale and a frequency ratio between false alarms and missed events."*

L179: Perhaps more a comment than anything else, but I would be very wary of applying a neighbourhood verification method on top of using a neighbourhood post-processing approach first.

*This is correct : as discussed in Schwartz and Sobash (2017) and Ben Bouallègue and Theis (2014), there are pros and cons to using neighbourhood in verification. We have investigated the relevance of this issue to our study in section 4.2 and demonstrated that, depending on user types, verification neighbourhood can make the forecasts look better, or not. Since the verification scale is ultimately a choice of the user (i.e., an assumption that we have to make about what the user wants) rather than a property of the forecasting system, it would be difficult to argue that applying neighbourhood in verification is right or wrong, in a general sense.*

L186 and L191: the 4- and 30-mm thresholds appear somewhat at random here. Is there any rationale for these values?

*(Also noted by the other referee) These intensities are arbitrary, as now stated in the text with a short explanation: "an arbitrary intensity threshold of 4~mm, which corresponds to fairly common rainfall in the studied area." and then "an arbitrary intensity threshold of 30~mm, which is reached much morerarely than 4~mm, but is frequent enough to produce robust statistics with our dataset."*

L219: I think you meant that it gives 4 times more weight to c than to b?

*Yes, this has been fixed.*

L220: I would replace non-detections with missed events

*Done*

L260: I feel this "only" is misplaced given that 0.12 < 0.3

*"Only" deleted*

L271: missing bracket

*Corrected*

L274: I think you mean Fig 3 here

*Yes, corrected*

L295: great illustration of the value of the FDR over POFD

*Thanks*

L317: I am not that familiar with the F2 score but we know that the ETS can be optimised by over-forecasting. It looks like the F2 can as well. Over-forecasting is by far the most useful mechanism for reducing missed events (of course at the expense of the false alarms!)

*Yes, that is what the text implies.*

L320: This is probably an important result and one worthy of elevating more in the abstract perhaps? For user H the ensemble probably has more value, even if the scores are lower.

*Inserted into the abstract as "The optimal threshold depends on the choice of forecast performance metric, and the superiority of the ensemble prediction over the deterministic control is more apparent at higher precipitation intensities."*

L332: This would be consistent with the notion that whilst spread is necessary, too much spread is not good.

*A corresponding remark has been added: " Larger dressing widths were found to degrade the scores, which suggests that increasing too much the ensemble spread would be detrimental."*

Figs 5, 6 and 7 are very small. The final version should have bigger panels. These figures contain some important findings.

*Panels in these figures (Fig 10, too) are now stacked vertically and scaled with full page width in order to improve readability.*

L343: What Fig 5b shows really well is how the use of some sort of neighbourhood improves the score.

*Yes*

L346-349: You'd hope that the ensemble (especially) can capture some of that spatial uncertainty. Not all heavy rainfall events are convective. Perhaps it would be better to say, that if 30 mm/6h are convection-driven, these come with large spatial uncertainties. To say that location errors affect user H more than L, is perhaps a little disingenuous because user H is a subset of user L. I strongly suspect that if user L was hit by the that 30 mm/6h they would still be affected by it!

*The presence of thunderstorm location uncertainties was a comment about our dataset, not a general one. This is now mentioned explicitly the last sentence is now corrected as "errors seem to affect user H \*proportionally\* more than user L" (since our scores are functions of error rates, not absolute error counts)*

L365-368: This is the double penalty effect in action and why we use upscaling.

*Comment added: "In both cases, forecast scores F2 and ETS strongly increase with verification scale, which corresponds to the "double penalty effect" mentioned in the introduction."*

L383: it might be useful to see what N is here for each of the thresholds

*This information is now broadly provided in a new paragraph: "In terms of sample size, one can note that in each period, the number of observation points that exceed threshold intensities (20, 40, 60, 80~mm) is of the order of (30000, 6000, 1400, 500). Since neighbouring data points are likely to have correlated error statistics, the actual number of independent data is likely to be at least one order of magnitude smaller, so that 80~mm can be regarded as an upper limit on the intensities at which our statistics are likely to be robust."*

L417: suggest longest instead of highest

*Corrected*

L435: What it does suggest is that prolonged periods without rain/events can be detrimental to aggregated statistics. It is therefore not the length of time that is important but the number of events in that time window.

*Comment inserted, thanks: " It was an exceptionally dry period, so this feature does not seem significant. It suggests that the number of events should be taken into account when defining the time intervals over which aggregated statistics are derived."*

Fig 10. I am not entirely sure I like the highest totals to the left. It is somewhat counterintuitive.

*The x-axis has been reversed as suggested*

L501: I guess what this example tells me is that the basic characteristics of the forecast (ensemble or deterministic) are still very much fundamental to the success of applying a user-relevant decision threshold. The outcome will only be as good as the forecast itself.

*Yes*

L516: This is a crucial point. Many countries are constrained in not issuing warning areas which are too large. Therefore, having specific and tight warning areas is highly desirable for many met services.

*Comment added: " The $q_H$-quantile detected the whole event at the price of a less precise framing of the area at risk, and of an underestimation of maximum intensities: answering is a matter of preference between detection,  false alarms, and peak intensity. The $q_H$-quantile produced many false alarms which may not be acceptable as a warning practice in many meteorological institutes."*

L528-531: Perhaps one of the good things about this way of post-processing the forecasts is that it does provide this alternative "views" of the same forecast.

*Comment added: "One can regard the $q_H$- and $q_L$-quantile maps as being two possible "views" of the same forecast."*

***Response to reviewer RC2: 'Comment on egusphere-2023-3111', Anonymous Referee #2, 07 May 2024, https://doi.org/10.5194/egusphere-2023-3111-RC2 (author's response below is typeset in italics)***
*We thank the reviewer for the thorough and very helpful comments.*

**Review to EGUsphere-2023-3111 "Probabilistic short-range forecasts of high precipitation events: optimal decision thresholds and predictability limits".**

In their manuscript, the authors present a new data based ensemble post-processing method to summarize ensemble forecasts as an input to e.g. high precipitation warnings. This new approach takes into account user requirements in terms of target rain intensities and tolerance to false alarms and event non-detection. The method is based on a statistical learning of forecasts archived over a few months. The output is a set of optimal decision thresholds expressed as an ensemble probability threshold or an optimal quantile level. To examine the sensitivity of the procedure, several impact experiments have been made and furthermore 3 real case studies have been presented.

The authors found out that ensemble predictions objectively outperform the corresponding deterministic forecasts at low precipitation intensities. Precipitation forecast skill generally decreases with intensity (expect for orographically enhanced events), while it increases with accumulation time, and verification scale. The post-processing method proposed in this manuscript can also be regarded as a kind of ensemble calibration since forecast biases are compensated.

The paper presents an interesting approach which could be helpful for human forecasters as a kind of supporting tool in case of severe weather but could also be included into automated forecast applications. The structure of the manuscript is clear and conclusive and it is written in excellent English supported by well-arranged figures (some of them should be enlarged) and tables. The article is addressing topical methods in probabilistic weather forecasts/warnings and hence fits very well into the scope of the journal.

However there are some things that should be addressed before final publication.

I recommend the paper is accepted with some minor revisions.

*We agree with the recommendation and respond point by point to the specific comments below.*

General: There are a lot of blanks in the text (before .,:) which should not be there. Also some inconsistencies (e.g. abbreviations, figure references, numbering, etc.) are present. Check the whole manuscript with respect to that.

*Most blanks before punctuation were intended to separate numbers from the following punctuation. As requested, they have now been deleted. The typesetting of figure references and of the bibliography has been rechecked and cleaned up.*

1. Page 2, L38: significance or skill measures. Typo.

*Corrected as "of"*

2. Page 3, L82: Reference to Wikipedia. Can you add a more scientific reference to this statement.

*Replaced by a reference to Nancy Chinchor (1992) who originally introduced the F-metric: "Chinchor, N.: MUC-4 evaluation metrics, In Proceedings of the Fourth Message Understanding Conference, McLean, Virginia, June 16-18, 22--29, 1992, https://aclanthology.org/M92-1002.pdf, last access: 27 May 2024, 1992."*

3. Page 3, L90: "prediction scales of the order of 30km, which is the resolution at which weather warnings are often issued in European countries. Add some reference for this statement.

*Reference to Legg and Mylne (2004) now repeated here, it provides the information for the UK. Current official weather warning maps are public knowledge (e.g. www.meteoalarm.org), although they do not appear to be documented by "scientific" references. The new text is "we will focus on prediction scales of the order of 30 km, which is the typical resolution at which weather warnings are often issued in European countries (Legg and Mylne 2004)."*

4. Page 4, L121: The following paper will be structured as follows. Missing word.

*Corrected*

5. Page 4, L122: Section 6 is described twice in this sentence.

*Corrected*

6. Page 5, L138: What do you mean by "upper boundary conditions"? upper air? Describe it.

*A reference to Descamps (2007) has been added. The numerics of the boundary coupling are extensively documented in Termonia et al (18) so this reference is repeated here. The new text is "Perturbed AROME-EPS members start from the deterministic initial state to which perturbations from an ensemble data assimilation system are added. They also use large-scale boundary conditions from the global PEARP ensemble prediction system of Météo France, as described in Descamps et al (2007) and Termonia et al (2018)."*

7. Page 7, L169: blank is used before : This is in the whole text, sometimes you have a blank before :, sometimes not. Be consistent.

*All blanks have now been deleted before dots, commas, colons, semicolons.*

8. Page 7, L186: is there a reference why you use 4mm and 30mm as threshold? Add reference or description.

*(Also noted by the other referee) These intensities are arbitrary, as now stated in the text with a short explanation. The new text is "We will characterize the corresponding forecast performance using the equitable threat score (ETS) with an arbitrary intensity threshold of 4~mm, which corresponds to fairly common rainfall in the studied area." and then "This objective will be modelled by the F2 score using neighbourhoods $R_f = R_o = 30$~km and an arbitrary intensity threshold of 30~mm, which is reached much more rarely than 4~mm, but is frequent enough to produce robust statistics with our dataset."*

9. Page 8, L198: produce. Typo.

*Corrected*

10. Page 9, L227: A word is missing in this sentence (e.g. of these terms).

*Corrected*

11. Page 10, L267: "should not be interpreted as a …" Missing word.

*Corrected*

12. Page 10, L271: Missing bracket.

*Corrected*

13. Page 10, L275: You mean Figure 3a – wrong reference to figure.

*Corrected*

14. Page 15 Figure 5: too small figure – should be enlarged.

*Done*

15. Page 16 Figure 6 and 7: too small figures – should be enlarged.

*Done*

16. Page 16, L396: This is strongly depending on the region, since seasonal precipitation characteristics are completely different between the northen parts of France and the Mediterranean region. This should be stressed more.

*Comment added as "As already implied by Fig. 1b, precipitation characteristics are completely different between the northern part of France and the Mediterranean region, which probably leads to variations in the performance scores."*

17. Page 17, L 413: This sentence is confusing – add 6h-accumulated precipitation…

*Corrected*

18. Page 17, L420: Why 12 to 15 local solar time? I do not understand, I thought you are using 21 UTC runs.

*Corrected, some text was on the wrong line. The new text is "Fig. 8 shows that the scores and statistics are nearly independent from range 9 to 36~h. There is a small decrease of less than 10\% in F2, ETS, hit rate and $p_{\mbox{opt}}$. A hint of diurnal cycle is visible. The lower predictability at ranges 15 to 18 (12 to 15 local solar time) is probably due to diurnal development of convection during the warm season."*

19. Page 18, section 6.2 As mentioned before, this seasonal dependency could be very different from region to region – a statement about this would be fine.

*Statement added as suggested: "This seasonal dependency could be very different from region to region."*

20. Page 19, L461: But this will cause a lot of false alarms.

*Added as suggested: " Heavy rain warnings for accumulations over 1 to 3~h need to be triggered at very low probability levels (10\% or less) because their hit rates are quite low, and it will cause a lot of false alarms (as illustrated by the FDR curve)."*

21. Page 20 Figure 10: too small figure – should be enlarged.

*Done*

22. Page 20, L475: and forecasters on duty do normally not have the time to look on each members separately.

*Comment added as suggested: " Forecasters on duty may not have the time to look at each member, which tends to be a tedious task:"*

23. Figures 11, 12, 13: UTC should be written in capital letters (in the complete manuscript).

*Done*

24. Page 20, L504: On the other hand you have an overestimation in the central parts of the domain in Figure 11 in qH quantile plot compared to the deterministic plot and the observations.

*This remark has been added in the text, although the issue was already suggested in the text. The new text is " Since the $q_H$-quantile overestimates precipitation in the central parts of the plotted area, a high precipitation warning based on $q_H$-quantile values above 50~mm would arguably result in many false alarms because of the limited sharpness of the ensemble forecast, but it nevertheless would still contain better information about the areas at risk than the deterministic forecast (Fig. 11d).*

25. Page 21, L506: can strongly depend. Remove the s in depends.

*Done*

26. Page 23, L515: It would be interesting to have a "classical" ensemble median or probability map in comparison to qL and qH quantiles plots for this case.

*Added as two extra panels in Fig.12, commented by a new sentence: "The $q_H$-quantile (Fig. 12d) successfully encompassed the observed high precipitation event with more accurate values. The deterministic forecast (Fig. 12d) was more precise than the $q_H$-quantile in terms of false alarm avoidance and maximum values, but its peak was located too far north, missing the observed peak by about 20~km. Both $q_L$- and $q_H$-quantiles are clearly superior to the raw ensemble median (Fig. 12e) and probability map of exceeding 75mm (Fig. 12f), both of which failed to convey the risk of damaging precipitation (which occurs beyond 100mm intensities in this area)."*

27. Page 23, L532: that in instead of than.

*Corrected*

**(END OF RESPONSE TO REVIEWERS)**